# Random-Access Accelerator (RAA): A Framework to Speed Up the Random-Access Procedure in 5G New Radio for IoT mMTC by Enabling Device-To-Device Communications

**DOI:** 10.3390/s20195485

**Published:** 2020-09-25

**Authors:** Abel Rodriguez Medel, Jose Marcos C. Brito

**Affiliations:** National Institute of Telecommunications INATEL, Av. João de Camargo, 510-Centro, Santa Rita do Sapucaí-MG 37540-000, Brazil; brito@inatel.br

**Keywords:** IoT, mMTC, random-access procedure, D2D, NR

## Abstract

Mobile networks have a great challenge by serving the expected billions of *Internet of Things* (IoT) devices in the upcoming years. Due to the limited simultaneous access in the mobile networks, the devices should compete between each other for resource allocation during a *Random-Access* procedure. This contention provokes a non-depreciable delay during the device’s registration because of the great number of collisions experienced. To overcome such a problem, a framework called *Random-Access Accelerator* (RAA) is proposed in this work, in order to speed up network access in *massive Machine Type Communication* (mMTC). RAA exploits *Device-To-Device* (D2D) communications, where devices with already assigned resources act like relays for the rest of devices trying to gain access in the network. The simulation results show an acceleration in the registration procedure of 99%, and a freed space of the allocated spectrum until 74% in comparison with the conventional *Random-Access* procedure. Besides, it preserves the same device’s energy consumption compared with legacy networks by using a custom version of *Bluetooth* as a wireless technology for D2D communications. The proposed framework can be taken into account for the standardization of mMTC in *Fifth-Generation-New Radio* (5G NR).

## 1. Introduction

For a device to share data with other devices, equipment, or infrastructure—which are not physically attached to it—wireless communication technologies have arisen that establish a connection between them by means of radiofrequency resources. Just as in wired communications, the wireless approach includes control data in every transmission to assure that the connection complies a required service quality or service level agreement, security, information integrity, authentication, and authorization. There are many protocols associated with the control data to dictate the way in which the communication is established and how the devices should behave to accomplish all the requirements. When a device transmits data to another device, the information data can be transmitted and received with minimal control data alongside. However, if a third device transmits data to the other two, their transmissions could collide delaying the communication, affecting real-time data processing and draining battery reserves faster in the case of portable devices. Therefore, the control algorithm must be enhanced to avoid two or more devices transmitting at the same time.

*Medium Access Control* (MAC) is one of the most important aspects of communication networks. It is in charge of the coordination of multiple entities sharing the same physical channel to avoid or reduce the collisions in it [1]. In the case of mobile networks, the MAC method employed is the *Mesh Coordination Function* (MCF), which is based on a *Contention Channel Random-Access* procedure where limited management frames are reserved for future transmissions. The neighboring devices listen to the reservations and do not transmit during the reserved periods [2]. Once the reserved period ends, the neighboring devices can each reserve a frame. Thus, it is possible that more than one device tries to make the same reservation at the same time provoking a collision and consequently not reserving a frame for either device. One more time, the devices should wait for the current reserved period to end. As the number of devices trying to reserve a frame increases, the probability of collisions also increases and the waiting time becomes longer [3]. In addition, the current networks employ a *back-off* algorithm to extend the waiting times of every device randomly [4]. That time can even be in the order of seconds, which helps avoid collisions, but increases the waiting time even more. Therefore, due to the exponential growth of connected stations, especially IoT devices [5], the mobile networks will face a difficult challenge handling the medium access to reduce the access time of the devices in the upcoming years.

In addition, the 5G network will use higher frequencies for communication in order to expand the available bandwidth to support massive communication scenarios like mMTC [6]. The use of such frequencies leads to more propagation losses and, as a consequence, the communication distance should be shorter to avoid signal degradation [7]. Therefore, the cell size will be smaller than in legacy networks, which increases the probability of a device crossing the limits of the cell. In the new cell where the device has camped, the device must perform the *Random-Access* procedure to register in the network again. This procedure is not efficient because there are limited vacancies to connect to the network. Only 64 vacancies (preambles) are periodically available for all the devices trying to gain access to the network [8]. In mMTC, the number of devices is much higher than the number of available preambles. Many of these devices would select the same preamble. Every preamble is associated with a channel. The devices that select the same preamble will send back the selected preamble to the gNB (5G Base Station) in the same channel, thus provoking collision [9]. The more devices trying to gain access at the same time, the higher the collision probability, the higher the access delay [10], and the higher the energy consumption.

The applications related to motion are the most affected by the discussed problem. The more mobility a device running these kinds of applications has, the more probability of crossing cell limits and a greater number of attempts for a device to have access. Some examples of those applications are factory automation, where a set of machines are in motion; autonomous driving, where cars are expected to cross the coverage area limits very often [11]; wearable devices that can be carried by a person along its way to notify the police about criminal incidents as rapes or assaults [12]; and fitness tracking devices that measure calories and heart rates along the athlete’s trajectory [13].

In 5G, the minimum unit of assigned resources per device is a *Resource Block* (RB). Every RB has 12 subcarriers [14] and each subcarrier has 14 symbols in *New Radio* (NR) [15]. Therefore, there are 12 × 14 = 168 symbols within an RB. In the case of the IoT devices, most of them are sensors, and they require a few symbols to transmit/receive their data. For example, an IoT sensor that checks the room temperature in the wide range 0 ^0^F to 100 ^0^F will need 7 bits to cover the 101 possibilities. If the device uses *Binary Phase-Shift Keying* (BPSK) modulation where every symbol is composed of 2 bits, the device needs to transmit 3 symbols to send its temperature measurements. Therefore, 168 symbols is an excessive amount for the IoT sensor demands. Thus, the gNB will allocate 165 unused symbols to the IoT sensor.

The D2D communications allow that near devices exchange their unused resources to other devices. For a device to transmit part of its resources to another device, both devices need to agree on the channel they will use for the resource’s exchange. Every D2D technology has its own procedure for the channel agreement. Some technologies reach an agreement more quickly, but incur more energy consumption; other technologies are slower during this procedure, but consume less energy. In addition, the agreement procedure is influenced by the number of other agreements procedures that happen at the same time and in the same location. If more than a pair of devices agree on the same channel, their transmissions will collide. The D2D technologies were originally designed to avoid collisions, which means that collisions exist but the transmission/reception is based in reattempts until successfulness. In massive communications, the D2D procedures for the channel agreement and successful communication are prohibitively delayed due to the high device’s concurrence. The delay implies more transmission attempts and consequently more energy consumption. Therefore, the current D2D technologies need to be readjusted to comply with both low latency channel agreement and low energy consumption during communication in the mMTC scenario.

In this work, RAA is proposed as a framework to improve the *Random-Access* procedure of the legacy networks. The main goal is to reduce the waiting times during reserved periods by more efficiently handling the number of accessing attempts in the network. In addition, RAA conserves the energy spent by the devices during the entire process in comparison with the traditional approach, and the network resources are shared more efficiently between all the devices than in legacy networks. The procedure integrates not only the communication between the devices and the *Base Station* (BS), but also the D2D communications. The devices with already assigned resources by the network will serve as relays for new devices trying to access the network. Thus, the registered devices use their allocated resources to forward the pretending reservations of the new devices to the BS. Therefore, most collisions occur in D2D communications, and not in *Device-to-Infrastructure* (D2I) communication. The technologies explored for D2D are *Bluetooth* and *Wi-Fi*. Both of them were tested using an app for each one to verify their suitability for the proposed framework.

The rest of the paper is organized as follows. In Section 2 are presented the related works to mitigate the access and resource’s management constraints in NR. In Section 3 are discussed the *Random-Access* procedure and its limitations in detail. Then, in Section 4 RAA is introduced with all its features. Section 5 presents some simulation scenarios where are employed RAA and some assessments. Last, the conclusions of this work are given.

## 2. Related Work

In order to resolve the limitations raised in the previous section, some research propose mechanisms based on D2D communications as an alternative technology to assist the access demand. The authors of [16] introduced a new algorithm for contention resolution, called *Binary Countdown*. This algorithm is executed after receiving the resource allocation for the transmission of the *Connection Request* message, and before sending the *Connection Request* message to the gNB, which avoids unnecessary uplink communication (D2I) if a preamble collision has occurred. Every device will generate a random sequence of 0 and 1 with a variable length depending on the network load. The generated sequence is shared between nearby devices by a *sidelink* (D2D communication). Then, every device checks its sequence and compares it with the sequences arriving from nearby devices. If the current element from a device’s sequence has the same priority as at least one element from the rest of the sequences, the device will continue with the contention resolution procedure. If the current element from a device’s sequence has the highest priority and the rest of the elements from the other sequences has the lowest priority, the device wins the contention. However, due to the coverage restriction of the D2D technologies [17], all the devices in the cell do not communicate with each other. Then, it is possible that more than one device wins the contention in different groups and select the same preamble. Therefore, this procedure is based on the best effort to let devices gain access in the least possible time by reducing the collision probability. The principal limitation of this solution is the high number of *sidelink* communications between the devices that want to transmit a preamble for resource allocation using the *traditional D2D wireless technologies*. This could drain the *device’s battery* faster and could increase the number of collisions in a conflictive spectrum space like the 2.4 GHz band. Besides, the gNB must transmit an *extra message* periodically informing the devices of the sequence’s length for the next contention, which increases the processing load of the network. Unlike the work presented in [16], with RAA there is no agreement between devices to select the device or devices that will execute the *Random-Access* procedure and the ones that will not do it. Therefore, RAA allows that all devices have the same priority during the access procedure. RAA does not employ the *traditional D2D wireless technologies*, instead, the proposed procedure uses a custom D2D technology based on the *Bluetooth* algorithm with an extra *back-off* to reduce the number of collisions between devices during the D2D communication.

Due to the minimum unit of assigned resources per device is an RB in 5G, the number of symbols a device needs could be less than the total of symbols an RB has. The rest of the symbols unused by the device could be delegated to another device that needs it. Thus, it is introduced the D2D communications for resource delegation in [18]. That work proposes a framework which is divided into two main algorithms, one of them is for the device that requests resources (called *Cellular User Equipment* C-UE), and the other one is for the device that offers resources (denominated *Device-To-Device User Equipment* D2D-UE). The approach consists of the gNB updating a list with all possible devices that need resources, and sharing the list with the providers (devices with already assigned resources). The list is different for every provider because it contains only devices in the vicinity of the provider. One of the constraints of this solution is the *extra memory* a D2D-UE needs to have to store exclusively all its C-UE neighbors. The D2D-UE also needs to communicate very frequently with the gNB to update the C-UE neighbor’s list, which could drain the *device’s battery* faster. In this scenario, there are many *sidelink* communications that could increase the number of collisions in a conflictive spectrum space like the 2.4 GHz band by using the *traditional D2D wireless technologies*. Besides, the gNB must transmit an *extra message* periodically to update the identification of the nearby devices requesting resources in the connected devices list, which increases the processing load of the network. Contrasting with the approach exposed in [18], RAA does not allow that nearby devices offer their unused resources, instead, only the gNB allocates the resources for each device. The devices with allocated resources are intermediaries between the devices looking for access and the gNB. The gNB receives the new device’s demands via the intermediary devices with allocated resources in the mobile network. Then, the resource allocation is done by the gNB over a trustable channel and not over the unlicensed spectrum channels, in order to avoid collisions and reduce the resource allocation time.

A D2D-based *Random-Access* technique is introduced in [9]. It transfers the possible access congestion between the devices and the network to the *sidelink* communications between nearby devices. The access delay is significantly reduced, but it is not conceived the possibility of *sharing resources* in case some devices have more allocated resources than needed.

In [19], the authors considered various resource allocation strategies to more effectively handle the access in the mobile network for different slices: *enhanced Mobile Broadband* (eMBB), *Ultra-Reliable Low Latency Communication* (URLLC), and mMTC. Unlike the work presented by the authors of [19], the RAA procedure does not differently handle the types of slices. Thus, the eMBB, URLLC, and mMTC slices are treated with the same priority level. The work in [10] introduces a 2-step *Random-Access* approach instead of the 4-step conventional procedure by sending both control and information data in the same message. Unlike the work presented in [10], RAA is executed in parallel with the original traditional 4-step *Random-Access* approach in 5G. The approaches presented in [10,19] are focused only on the access blocking probability. They do not show the *elapsed time* for all device’s registration and there is no consideration of the *energy consumption* during the slicing procedure. Contrasting with those approaches, the RAA procedure presented in this work assesses the *elapsed time* for all device’s registration and considers the *energy consumption* of the devices and the gNB during the access procedure. In addition, the approach in [10] can negatively influence the device’s energy consumption because—in case of collision—the devices spend both control and information data energy in infructuous transmissions.

RAA exploits the D2D communications to achieve low latency access to the network. A new *back-off* was added to the D2D technologies used by the proposed framework in order to avoid collisions and accelerate the discovery procedure between nearby devices. The devices that want to be registered in the mobile network, also called requesters, send their resource’s demands to the nearby devices with allocated resources, named relays. The relays are forwarding devices; they retransmit the requester’s demands to the gNB. Then, the gNB handles two flows of resource requests: one from the traditional *Random-Access* procedure and the other from the RAA procedure. In the last case, the message containing the resource’s demands from the new devices includes the number of subcarriers and symbols needed for the devices to transmit/receive data. With this information, the gNB fits the exact demand into the spectrum and time resources. Therefore, the gNB can manage resource allocation better than traditionally. With RAA, the elapsed time during registration is reduced by 99% in comparison to the traditional *Random-Access* approach. Due to the fast access experienced by the devices, the device’s energy consumption remains the same as in the traditional *Random-Access* procedure.

## 3. Nr Random-Access Procedure

The *Random-Access* procedure in the 5G New Radio network is summarized in Figure 1. It starts when the gNB broadcasts the *Physical Random-Access Channel* (PRACH) configuration to the devices attempting to connect to the network. The configuration message is part of the *System Information Block 1* (SIB1). Preambles for resource allocation and an access probability are sent within the configuration message. The devices that receive this message execute the *Access Class Barring* (ACB) algorithm. ACB consists of the devices generating a random number between 0 and 1 [4]. If the generated number is equal or smaller than the access probability sent by the gNB, the devices can access the network. Once the devices are allowed to access the network, they select one of the preambles sent by the gNB in the PRACH configuration message [20]. The devices transmit the selected preamble to the gNB in a message called *MSG1*. After the gNB receives *MSG1*, the gNB broadcasts a message (*MSG2*) in response to the preamble transmissions. This message contains resource allocation for the transmission of the *Connection Request* message (*MSG3*). Then, the devices send *MSG3* to the gNB. The gNB sends a *Connection Setup* message (*MSG4*) in response to the connection requests including allocated resources for the devices transmit its data. *MSG4* acts like a contention resolution message. It assigns a 40-bit *Identifier* (ID) to identify only one device from a group of devices that both selected the same preamble and transmitted at different times. In that case, the gNB receives multiple requests for the same preamble but the gNB only replies to one device [21].

### Random-Access Procedure Constraints

In 5G, there are only 64 available preambles to reserve resources. The limited preamble number is a consequence of the use of a *Zadoff–Chu* (ZC) signal generator. The ZC generates orthogonal preambles with zero correlation, which avoids inter-signal interference. However, the generation process is difficult to perform in real-time and requires a large amount of memory to store the sequences [22,23]. Therefore, the shortage of preambles and the great number of devices derivate in very long periods of blackouts (no connection with the network) and many missed transmission opportunities.

After checking SIB1 information, devices know all preambles they can get for resource allocation. All devices select one of the preambles and send it to the gNB to request resources. Then, if more than one device selects the same preamble, they transmit in the frequency associated with the selected preamble. If both devices also transmit at the same time, their transmissions will collide because they are using the same channel [24]. However, collision is not detected yet. The devices that sent their preambles await for a *Random-Access Response* (RAR) during a *Random-Access Response Window* (*ra-ResponseWindow*). If no responses arrive in the *ra-ResponseWindow* period, the devices know a collision has occurred [25].

The number of collisions is reduced when the ACB algorithm is executed. ACB limits the number of simultaneous access attempts from devices that want to connect to the network. In this case, the devices use two types of information sent by the gNB within the SIB1 message to execute the ACB procedure: Barring rates *P_ACB_* ∈ {0.05, 0.1, …, 0.3, 0.4, …, 0.7, 0.75, 0.8, …, 0.95}, and barring times *T_ACB_*∈ {4, 8, 16, …, 512 s}. Then, every device determines its barring status. The devices pick their corresponding *P_ACB_* and *T_ACB_* from the lists above based on their classes (the class a device belongs is not important at this point). The devices generate a random number *g* = *U*[0, 1). If *g*≤ *P_ACB_*, the devices transmit a selected preamble; otherwise, the devices wait for a random time (*back-off*) calculated as *T_barring_* = [0.7 + 0.6 *U*[0, 1)]*T_ACB_* [4].

It is not difficult to realize that there is not a negligible waste of time when devices do not meet *g* ≤ *P_ACB_*, and therefore a lot of data transmission opportunities are missed. Let us check the amount of data that could be transmitted in the *back-off* period.

If a device does not meet the requirement to transmit a preamble, and it gets the minimum values from *P_ACB_* and *T_ACB_* (best case from the example above), the *T_barring_* = [0.7 + 0.6 × 0.05] × 4 = 2.92 ms. This time is equivalent to (2.92 ms/66.67 μs) = 43,798 symbols for 15 kHz numerology (numerology with longer symbol duration). Considering BPSK modulation (modulation with a minimum of bits per symbol), the total amount of data could be transmitted in the blackout period is 43,798 (№ symbols) × 1 (№ bits – BPSK) = 43,789 bits. Considering 64 *Quadrature Amplitude Modulation* (64QAM) modulation, the total amount of data transmitted is 43,798 (№ symbols) × 6 (№ bits – 64QAM) = 262,734 bits.

For the worst case of the example above, *T_barring_* = [0.7 + 0.6 × 0.95] × 512 = 650.240 ms. This time is equivalent to (650.240 ms/66.67 μs) = 9,753,113 symbols for 15 kHz numerology. Considering BPSK modulation, the total amount of data could be transmitted in the blackout period is 9,753,113 (№ symbols) × 1 (№ bits – BPSK) = 9,753,113 bits. Considering 64QAM modulation, the total amount of data could be transmitted is 9,753,113 (№ symbols) × 6 (№ bits – 64QAM) = 58,518,678 bits.

When devices meet *g* ≤ *P_ACB_*, their preamble transmission still could collide because all the devices that want to get resources from the network will acquire one of only 64 available preambles in NR. Therefore, ACB only alleviates congestion, it does not remove it completely. Thus, ACB will cause a negligible effect by reducing the number of devices contending for resources in mMTC.

## 4. Proposed Framework: Raa

The main purpose of RAA is that new devices (requesters) entering the cell coverage area discover at least one nearby peer with uplink grants that serve as a bridge between them and the gNB. If the new devices find another device that is registered in the cell that they want to have access to, the registered device can act as a relay for them to forward their resource requirements directly to the gNB. This procedure means that the new devices do not need to wait for the transmission of SIB1 by the gNB to start the traditional *Random-Access* procedure.

### 4.1. Raa Details

RAA is based on the *Best-Effort* paradigm. It always tries that a device requiring resources to transmit/receive data finds the first relay in the shorter possible time. The behavior of RAA is summarized in Figure 2.

Figure 2a shows the requester’s behavior. When the requester enters a cell coverage area, it tries to synchronize with the gNB downlink. To do that, the requester waits to receive the *Synchronization Signal/PBCH Block* (SSB), which is a message broadcasted periodically by the gNB. SSB contains the *Primary Synchronization Sequence* (PSS) and the *Secondary Synchronization Sequence* (SSS) signals. The requester extracts the *Sector ID* (SID) of the cell from PSS, and the *Group ID* (GID) of the cell from SSS [26]. Then, the requester computes the *cell ID* with both SID and GID and starts performing the proposed RAA procedure.

The requester also extracts the *Master Information Block* (MIB) from SSB. MIB provides the bandwidth of the downlink, the frame numbers, and the SIB1 location in frequency and time domain [27]. Then, the device scans for SIB1. Once the requester receives SIB1, it extracts the configuration parameters from the message to perform the *Random-Access* procedure.

At this point, the requester executes two procedures in parallel, the traditional *Random-Access* procedure, and the proposed RAA procedure. In the last case, the device starts looking for nearby devices that have already allocated resources—also called relays—in the same cell the requester is. For the discovery process, the requester broadcasts a discovery message via D2D communication, which contains the *cell ID* of the cell the requester has camped, the number of subcarriers the requester needs for downlink (SC_DL_), the number of subcarriers the requester needs for uplink (SC_UL_), the number of symbols the requester needs for downlink (NoSym_DL_), and the number of symbols the requester needs for uplink (NoSym_UL_). The requester sends the discovery message periodically during an inquiry interval. The requester stops the discovery if it receives either an acknowledge message from a nearby relay or a RAR from the gNB. If neither of those messages arrives, the requester continues discovering relays until the inquiry interval ends. After that, the requester applies a random *back-off* to avoid collision with other requesters that could be transmitting discovery messages at the same time. Once the *back-off* expires, the requester starts discovering relays again.

In case the requester receives an acknowledge message from a relay, the requester waits for a RAR from the gNB. Once the requester receives the RAR due to a previous relay discovery or as part of the traditional *Random-Access* procedure, the requester extracts from the RAR the information with frequency and time domain to transmit the *Radio Resource Control* (RRC) *Connection Request* message. Then, the requester waits for receiving the *RRC Connection Setup* message with the allocated resources. Now, the requester becomes a relay.

In Figure 2b is depicted the relay behavior. The relay listens to discovery messages during a scanning interval. If the scanning interval ends, the relay waits a long time to start scanning again. If during the scanning interval the relay receives a discovery message from a nearby requester, it will send an acknowledge message to the requester to notify that a relay has been found. The relay also forwards the requester message with the requester resource requirements to the gNB in the relay allocated uplink resources.

Figure 2c shows the gNB functionalities during the proposed RAA procedure. The gNB receives the information with the required resources for the requester via relay. Then, the gNB looks in a resource allocation table if there are available resources for the requester. If there are resources, the gNB informs the requester of resource availability via RAR. Otherwise, the gNB ignores the forwarded message.

### 4.2. Medium Access Control

The D2D communication between the requesters and the relays is performed through wireless communications. In this kind of communication, the medium access control cannot remove collision when more than one device transmits at the same time in the same frequency. Instead, wireless technologies are focused on collision avoidance. Thus, collisions could exist but not permanently.

RAA was conceived to be based on *Bluetooth* and *Wi-Fi* as the wireless technologies for D2D communication. However, the two technologies were tested 10 times in two developed Android apps that can be found here: https://github.com/Abel1027/D2D-Test-Apps.git. The app’s performances show that *Bluetooth* was 6.5 s faster on average than *W-Fi* during the device’s discovery. Therefore, *Bluetooth* is considered the D2D technology for RAA. In this case, requesters and relays select one of 32 available frequencies from the 2.4 GHz band to transmit the discovery messages and to listen to the discovery messages, respectively. If two or more nearby requesters select the same frequency and transmit their discovery messages at the same time, their transmissions will collide. Therefore, neither of the requesters will find a relay. However, the requesters select periodically new frequencies for the next transmissions within an inquiring interval. The only way that more than one requester selects the same frequency all the time is if they use the same stage of the *Bluetooth* internal 28-bit frequency generator clock. Therefore, an additional *back-off* is performed after an inquiring interval. This assures that the next time the requesters starts the discovery, they use different stages of the 28-bit clock to generate different frequencies for their transmissions. The same procedure applies for relays when they send back the acknowledge message to the requesters.

## 5. Simulation

RAA has been simulated using the *Python* programming language and the *SimPy* module [28]. *SimPy* is a discrete-time simulation package. Why is the module selected for the simulation? Why is it not used as a continuous-time simulation tool? The answer is related to the computation capabilities of the computers where the simulation could be executed and the time resolution a programming language can offer. On one hand, the simulation involves hundreds of simultaneous processes or threads. This can reduce the overall performance of the simulation by delaying some processes more than others. In a real scenario, every device performs its functions and neither of their processes is affected by other device’s processes. On the other hand, the minimum time resolution of the programming languages is in the order of the milliseconds and this does not satisfy the RAA environment, where the minimum time resolution is in the order of the microseconds. To overcome those problems, *SimPy* was taken into account. *SimPy* waits that all simultaneous loops within different processes finish, and then it saves every output of those processes with the same timestamp. This means that it does not matter if one process is faster than others, *SimPy* always waits for the slowest process and assigns the same timestamp as the faster. The time resolution is solved too because *SimPy* only saves timestamps as a float value, and not as a real-time value. For example, if *SimPy* is set with a time resolution of 1, *SimPy* interprets it just as 1 and not as 1 μs, or 1 ms. The time resolution is interpreted by the application and not by *SimPy*, which is helpful for process synchronization.

The simulation involves the functions performed by the gNB for device’s registration, the procedures executed by the devices to connect to the mobile network, and the relay functionality when the devices obtain resources. All these functionalities are summarized in Table 1.

### 5.1. Simulation Parameters

The gNB provides a set of parameters for device’s registration within MIB and SIB1. These parameters are not fixed and can vary in dependence of many factors such as the mobile network capacity, the number of devices attempting to connect to the network, the number of devices unregistered successfully from the network, and many others. Simulating all the different parameters is a very complex task. Thus, the simulation is based on the basic parameters offered by the network. For example, if all possible values from the *ra-ResponseWindow* are {*sl1*, …, *sl80*}, the simulation only selects the first one (*sl1*). Table 2 and Table 3 summarize the parameters used in the simulation by the gNB and the devices, respectively. Table 4 describes the metrics used in the simulation.

### 5.2. Simulation Results

In this subsection, we present the results for five simulation scenarios where a number of devices try to connect to the network and there are no devices registered yet. The first scenario simulates the traditional *Random-Access* procedure without using the proposed RAA procedure. The other four scenarios were simulated using a customized *Random-Access* procedure and RAA at the same time. The new *Random-Access* procedure consists of the devices expanding the limits of a list where a new random value is selected every time a device is expecting a *Random-Access Response* and the *ra-ResponseWindow* expires. For example, a requester device sends a *Random-Access Request* and it waits for a response during *ra-ResponseWindow*. If that interval expires and there are no received responses, the device selects a random value from the list [0, 1]. This value represents the number of subsequent SIB1s that the device will not listen to. After that, the device can listen to incoming SIB1s and executes the *Random-Access* procedure again. If the device does not receive a *Random-Access Response* again, the list becomes [0, 1, 2] to ensure that the device could be delayed another SIB1 period to avoid collisions in a chaotic scenario.

The difference between the last four scenarios is the wireless technology used and the frequency generation in the 2.4 GHz band for the discovery message transmissions. The second and third scenarios involve RAA using two *Bluetooth* algorithms: One using the internal 28-bit clock in every device to generate the transmission frequency, and the other generating the frequency randomly. The fourth and fifth scenarios simulate RAA using a customized version of *Wi-Fi* for both the 28-bit clock frequency generator and the random frequency generator. *Wi-Fi* is referred to as the classic *Wi-Fi* with its respective transmission power but it incorporates the *Bluetooth* algorithms.

The last four simulations commented above were tested in another four circumstances. In the first one, the requester devices start the RAA functions after receiving the SIB1, and in the second one the requester devices start these functions since the beginning (before receiving SIB1) when the devices want to connect to the network. The last two circumstances are a variation of the second and third circumstances where the gNB selects the frequency that every registered device (relay) will use to listen to incoming discovery messages from remote requesters.

Figure 3 shows the four circumstances for the total energy spent by a specific number of devices that want to register in the mobile network. Besides, Figure 4 shows the time spent until the last device is registered for the same circumstances. The figures are related to the scenario where is used *Wi-Fi* as wireless technology and the 28-bit clock for frequency generation. The simulation shows the results for 100, 200, 300, 400, 500, 600, 700, 800, 900, and 1000 devices attempting to acquire resources from the network. Looking at these figures is obvious that the best circumstance—less energy consumption and less waiting time during the device’s registration—is the RAA procedure starting before the reception of the first SIB1. The best circumstance is also selected in the rest of the scenarios. Once the best circumstances from every scenario are selected, they are compared with each other in terms of number of collisions, energy, and elapsed time during registration.

Note that the results of the simulations are the average of ten independent simulations. For each one of the individual simulations, it was employed a different seed for randomization. The seeds are in the integer range [0–9] for each simulation, respectively.

#### 5.2.1. Collision Analysis

When RAA is used, it is expected that the total number of collisions in the 2.4 GHz band is higher for *Wi-Fi* than for *Bluetooth* because of the wider coverage range in *Wi-Fi*. However, the simulation results demonstrated that in the *Wi-Fi* scenario, where the random frequency generator is used, fewer collisions occur than in the *Bluetooth* scenarios. This is because this *Wi-Fi*-based scenario registers all the devices in a shorter period of time compared with the others. The other *Wi-Fi* case (28-bit clock) is not fast enough during devices registration and cannot reach a smaller number of collisions. In the mobile network band, the two *Wi-Fi* cases experiment fewer collisions than *Bluetooth*. This happens for the same reason that was discussed before in the 2.4 GHz band. The *Wi-Fi* cases are faster registering devices than the collision per time unit rate in this band. Figure 5 shows the total collisions in the two bands: the 2.4 GHz band and the mobile network band for the two *Wi-Fi* and *Bluetooth* scenarios. From the figure, the *Wi-Fi* case that uses the random frequency generator is the best case. It experiences a smaller number of collisions in comparison with the other RAA scenarios. However, this best case involves a very high number of collisions compared with the classic *Random-Access* procedure.

#### 5.2.2. Energy Analysis

The total energy spent in the 2.4 GHz band is higher in the *Wi-Fi* cases as expected because of the higher transmission power associated with this wireless technology. Although the *Wi-Fi* case—where the random frequency generator is used— experiences a smaller number of collisions in this band than the *Bluetooth* cases, it spends more energy than the *Bluetooth* cases. This is possible because the energy spent in the *Wi-Fi* cases is 16 times the energy spent by *Bluetooth*, which does not compensate the gain experienced by the *Wi-Fi* case about the number of collisions in the 2.4 GHz band. Unlike the energy spent in the 2.4 GHz band, the energy consumption in the mobile network band is more correlated with the number of collisions experienced for every scenario. The similitude is due to the same power transmission used in all RAA procedures when the devices are communicating with the gNB in the mobile network band. In this case, the devices use the same power regardless of the D2D technology. Therefore, a smaller number of collisions in the mobile network band means less energy consumption in this band. Figure 6 shows the total energy spent by all devices during registration in the two bands: the 2.4 GHz band and the mobile network band for the two *Wi-Fi* and *Bluetooth* scenarios. From this figure, the *Bluetooth* cases experience lower energy consumption. Although the *Wi-Fi* cases are faster for device’s registration, the energy spent in every transmission far exceeds (16 times) the energy that the *Bluetooth* approach consumes.

Figure 7 shows the energy spent by the gNB in every scenario. This figure infers that RAA is much faster than the classic *Random-Access* procedure because it is expected that the energy spent by the gNB remains almost constant for every procedure. Then, if there is a great difference between the two procedures about energy consumption by the gNB, is because of the great difference about the elapsed time for device’s registration in every procedure. In all scenarios, the gNB transmits periodically the same amount of information. However, there is a small variation for these scenarios because of the number of responses the gNB sends to the requester devices. The energy consumption of the gNB depends on the number of collisions and the elapsed time for all device’s registration. More collisions and more delays in device’s registration mean that the gNB will receive more *Random-Access Requests* and the gNB will send more *Random-Access Responses*, which incurs in more energy consumption.

#### 5.2.3. Time Analysis

From Figure 8, it can be seen that the elapsed time for all devices registration when are used the RAA procedures overcomes the classic *Random-Access* procedure by far. The elapsed time for all the RAA scenarios is always approximately 100 ms or less; meanwhile, the elapsed time for the classic *Random-Access* procedure is always above 10 s. This great difference is because of the ACB algorithm used in the classic *Random-Access* procedure. On one hand, the ACB back-offs many attempts of device’s registration in the order of seconds, causing that many of these devices register in the network in very distanced intervals, as it is depicted in Figure 9a. On the other hand, the RAA approach redistributes the device’s registration more regularly in time, meaning that there are no large periods of isolation between groups of devices, see Figure 9b.

#### 5.2.4. Resource Allocation

Figure 10 shows the number of allocated resources for the classic *Random-Access* procedure and the RAA approach. In the x-axis are depicted the resource’s demands for 9 cases. For example, the first one (12/5) means that all the devices request 12 subcarriers and 5 symbols for their data reception/transmission. The y-axis represents the total number of subcarriers offered by the gNB. The allocation process is done by the gNB, placing the resources offered to every device alongside other device’s resources until the 14 symbols of a subcarrier are occupied. From the figure, it can be seen that the gNB allocates the requested resources with more flexibility when it is used the proposed RAA procedure. In the RAA case, the devices only request the resources they need. In the classic *Random-Access* procedure, the devices only ask for resources but they do not notify the exact number of subcarriers and symbols they need; therefore, the gNB assigns an entire RB to every device. The resource allocation made by the use of RAA overcomes the classic procedure especially when the devices that want resources from the mobile network are IoT devices. These kinds of devices require a small number of symbols for their transmissions. They are expected to request between 1 and 5 symbols, and less than 12 subcarriers. In that case, the RAA approach overcomes the classic resource allocation procedure in 30% of cases for the simulation results of 100 devices requesting resources (Figure 10a,b). When 1000 devices attempt to obtain resources, the resource allocation procedure using RAA overcomes the classic procedure in 74% (Figure 10c,d).

#### 5.2.5. A More Real Scenario

All the above simulations were made exclusively in this work, and they are available on GitHub: https://github.com/Abel1027/Framework-To-Speed-Up-RACH/tree/master/Framework%20Simulation. As they represent the scenario where there are no registered devices in the mobile network at the time all devices arrive, it is interesting to simulate a more real scenario. A more realistic situation concerns a group of devices connected to the gNB and another set of devices, frequently smaller than the connected ones, trying to connect to the network. In the simulation, the same five cases discussed before were used (RACH and the best four cases of the proposed RAA procedure) with a group of 1000 connected devices. This means that, when the first requester device attempting to connect to the network is turned on (gets into the mobile network), there are 1000 devices that can serve as a relay for it. However, the requester device will camp in the coverage area of a subgroup of connected devices because they are distributed randomly along the cell coverage area. Figure 11 and Figure 12 show the results for the total energy consumption and the elapsed time for all device’s registration in every scenario. These simulations are found on GitHub too: https://github.com/Abel1027/Framework-To-Speed-Up-RACH/tree/master/Framework%20Simulation%20(1000%20Connected%20Devices%20at%20the%20Begining).

In this case, the total energy consumption decreases approximately 7 times compared with the scenario where there are not connected devices. The energy consumption for both *Bluetooth* scenarios is almost identical to the classic *Random-Access* scenario. The elapsed time for all device’s registration decreases too. Note that only 1000 devices are connected before the requester devices start trying to connect to the network. The density of devices in a real NR cell is much higher than a few thousand devices, and the energy spent by the registering devices is expected to decrease even more than the classic *Random-Access* procedure.

## 6. Conclusions

In NR, every time a device camps inside the coverage area of a cell or a device moves on from an LTE cell to an NR cell, it has to perform a *Random-Access* procedure to obtain resources from the network. During the execution, the device competes with other devices that are also requesting resources. This fight becomes harder when the number of devices contending for resource allocation is high because there are limited access opportunities, especially in mMTC scenarios. Therefore, the total time for device’s registration increases too much.

In this work, RAA was proposed as a framework that enables D2D communications to transmit and receive resource requirements messages between the devices that want to be registered in the mobile network and the devices that have already allocated resources. The registered devices act as relays and forward all the resources necessities of the no-connected devices to the gNB. For the D2D communications, four different customized technologies were used for device discovery and data transmission/reception: *Bluetooth* using the classic 28-bit clock for frequency generation, *Bluetooth* using a random frequency selector, and the same two approaches but using the *Wi-Fi* transmission power. The four technologies are integrated into the RAA procedure resulting in four RAA alternatives. All the RAA alternative’s performances were compared with the traditional *Random-Access* procedure for two cases: first, a bunch of devices starts looking for access to the mobile network when there are not connected devices yet, and second, the same number of devices try to gain access to the network but there are already 1000 devices with allocated resources. In the first case, the number of collisions experienced by the devices in the unlicensed band and the mobile network band was higher for the four RAA procedures than the number of collisions experienced by the devices in the same bands using the traditional *Random-Access* procedure. The energy spent by the devices was also higher for the RAA procedures. However, the energy consumption of the gNB is lower for the RAA procedures than the traditional *Random-Access* procedure. In the second case, where 1000 devices are acting as relays, the energy consumption of the devices is still higher than the traditional *Random-Access* procedure when the two RAA procedures with the *Wi-Fi* transmission power are used. However, if the RAA procedures use the *Bluetooth* transmission power, the energy consumption of the devices is the same as the traditional *Random-Access* procedure. In both cases, the four RAA procedures reduce the elapsed time for the device’s registration by 99% in comparison to the traditional *Random-Access* approach. The comparison results also show that the number of devices registered by time unit is more regular when the four RRA procedures are used than the traditional *Random-Access* procedure. In the traditional *Random-Access* approach, most of the devices register at the beginning of the access procedure, and then the number of devices registered per time unit is reduced proportionally to the elapsed time. The RAA procedures also overcome the traditional *Random-Access* procedure during resource allocation because RAA places every resource’s demand in the first empty space it fits from the resources grid. In this aspect, RAA overcomes the traditional procedure in more than 74% if the registered devices are IoT based.

The energy consumption related to the individual processing of each device has not been studied in this paper. It will be analyzed in future work. However, it is expected that the processing energy spent on every device will be low enough to preserve the required life cycle of the device battery. It was checked that with only 1000 connected devices acting like relays and distributed randomly, RAA incurs in the same energy consumption for the *Bluetooth* case like in the *Random-Access* procedure. Therefore, if the gNB commands that 1000 different devices act like relays every pre-established time in a super-populated NR cell, the low energy requirement for IoT devices will be fulfilled because just a very small number of devices is processing incoming data from nearby devices.

## Figures and Tables

**Figure 1 sensors-20-05485-f001:**
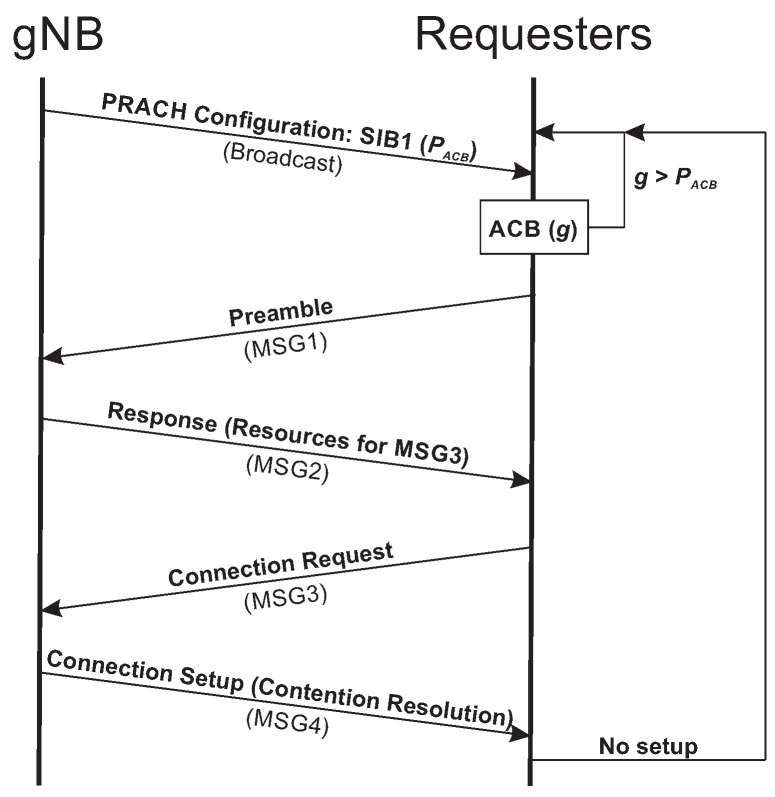
*New Radio* (NR) Random-Access procedure.

**Figure 2 sensors-20-05485-f002:**
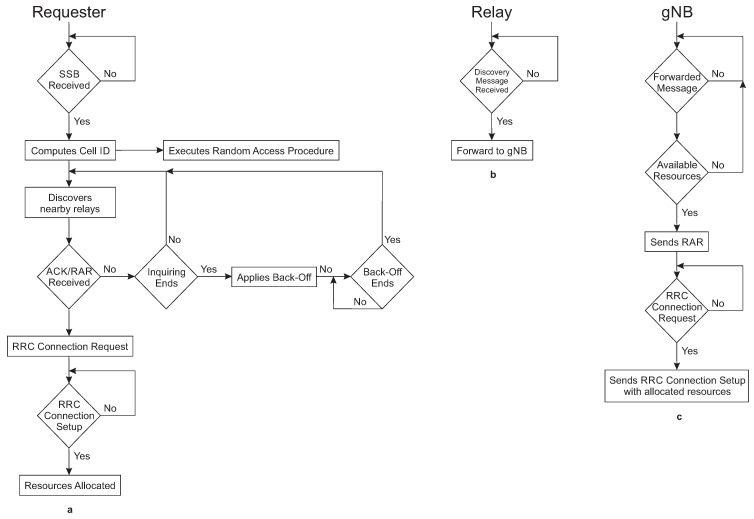
(**a**) Requester, (**b**) Relay, and (**c**) gNB summarized behavior pseudocodes.

**Figure 3 sensors-20-05485-f003:**
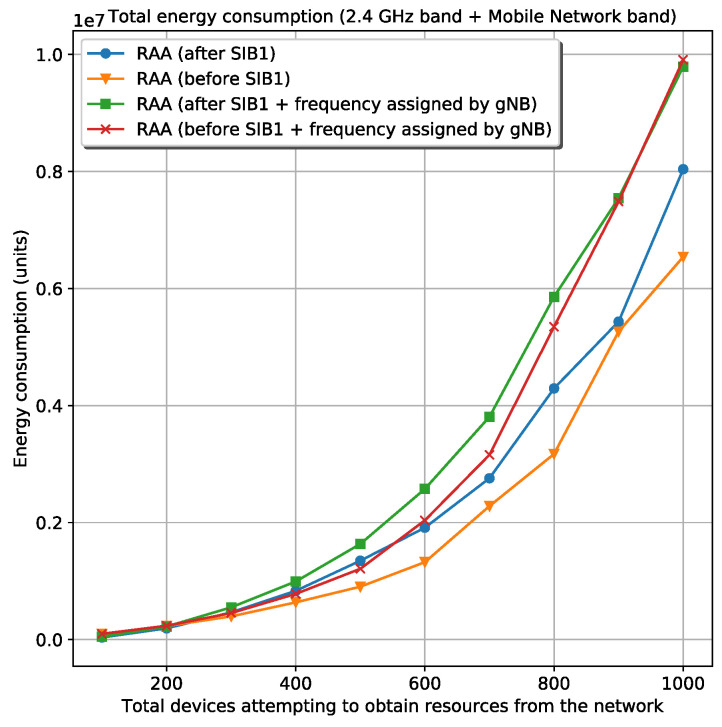
Total energy spent by all the devices attempting to obtain resources from the network in *Wi-Fi* using the 28-bit clock frequency generator.

**Figure 4 sensors-20-05485-f004:**
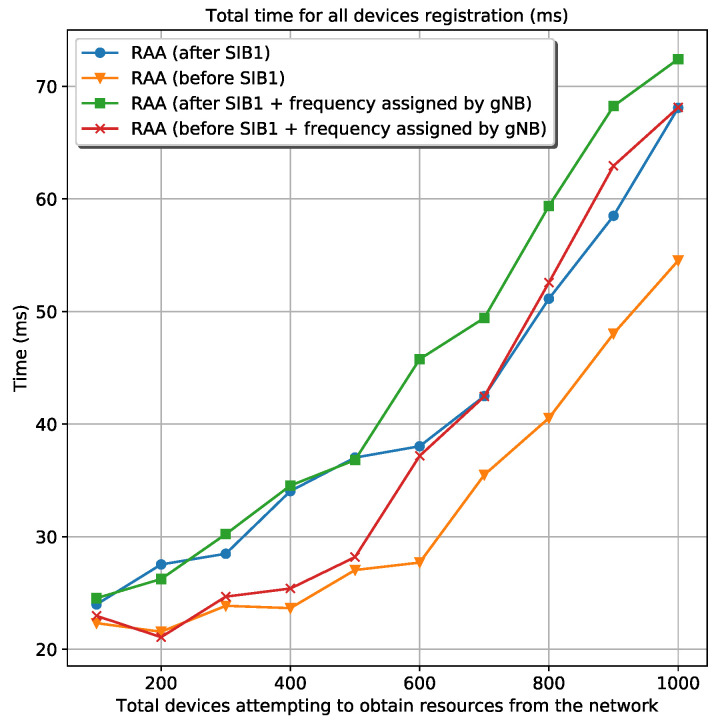
Total time spent by all the devices attempting to obtain resources from the network in *Wi-Fi* using the 28-bit clock frequency generator.

**Figure 5 sensors-20-05485-f005:**
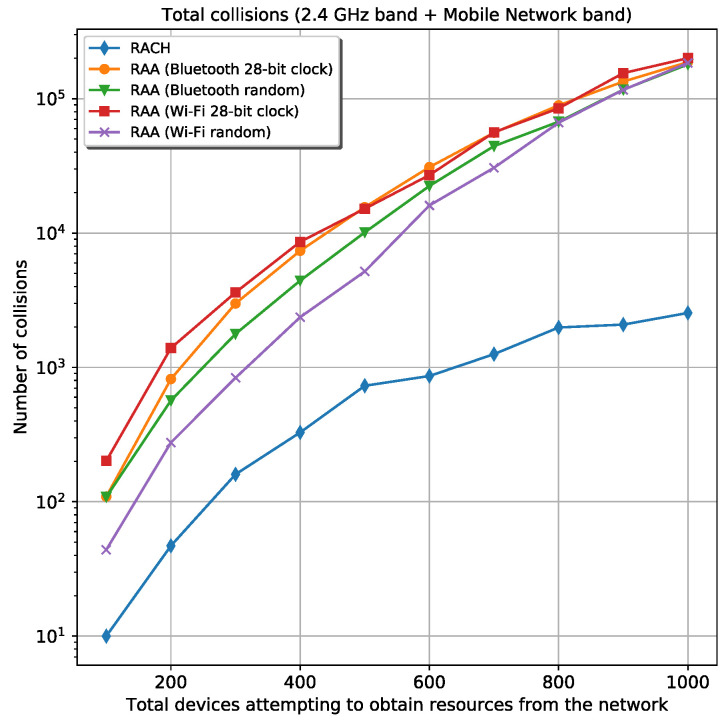
Total number of collisions in all bands for the classic *Random-Access Channel* (RACH) procedure and four different *Random-Access Accelerator* (RAA) procedures.

**Figure 6 sensors-20-05485-f006:**
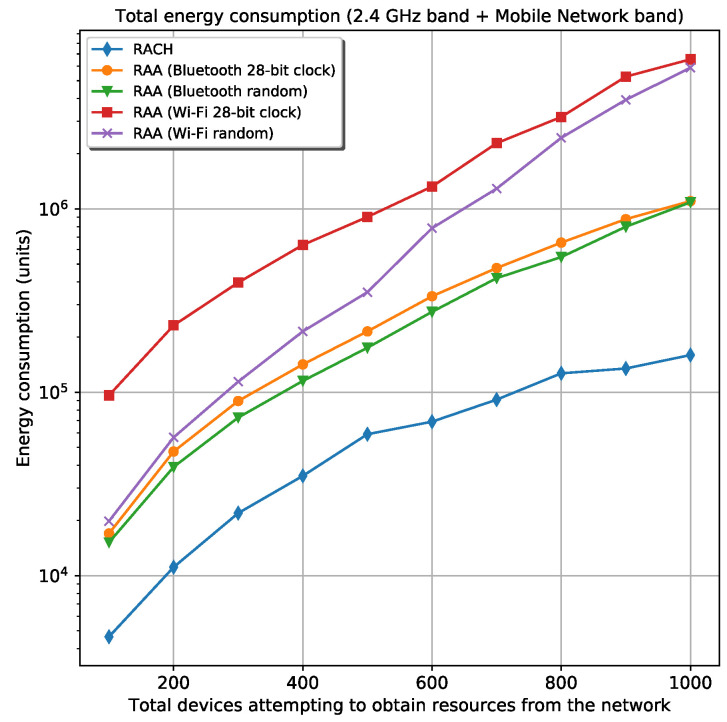
Total energy spent in all bands for the classic RACH and four different RAA procedures.

**Figure 7 sensors-20-05485-f007:**
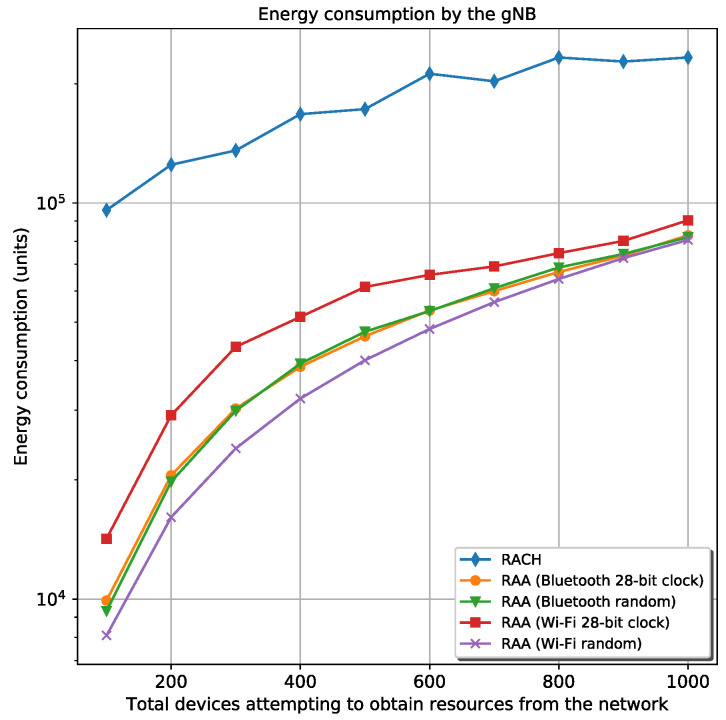
Total energy spent by the gNB for the classic RACH and four different RAA procedures.

**Figure 8 sensors-20-05485-f008:**
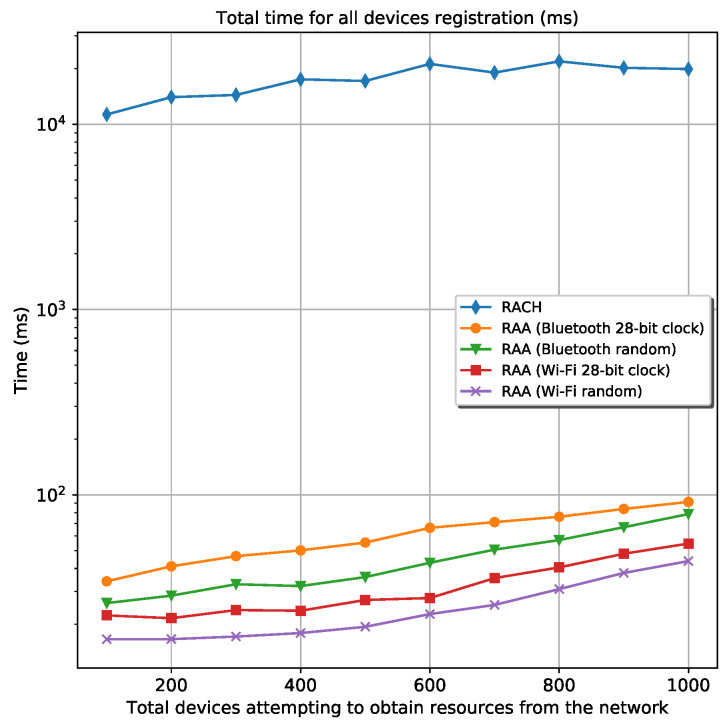
Elapsed time for all devices registration for the classic RACH and four different RAA procedures.

**Figure 9 sensors-20-05485-f009:**
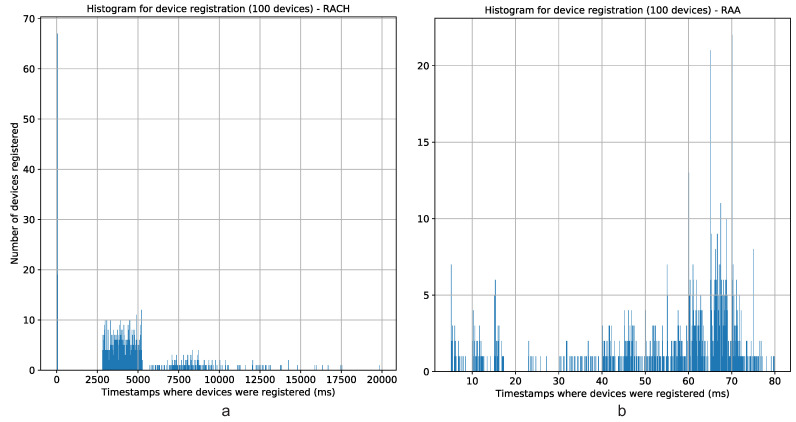
Number of devices registered per time unit for the (**a**) classic *Random-Access* procedure and the (**b**) RAA procedure.

**Figure 10 sensors-20-05485-f010:**
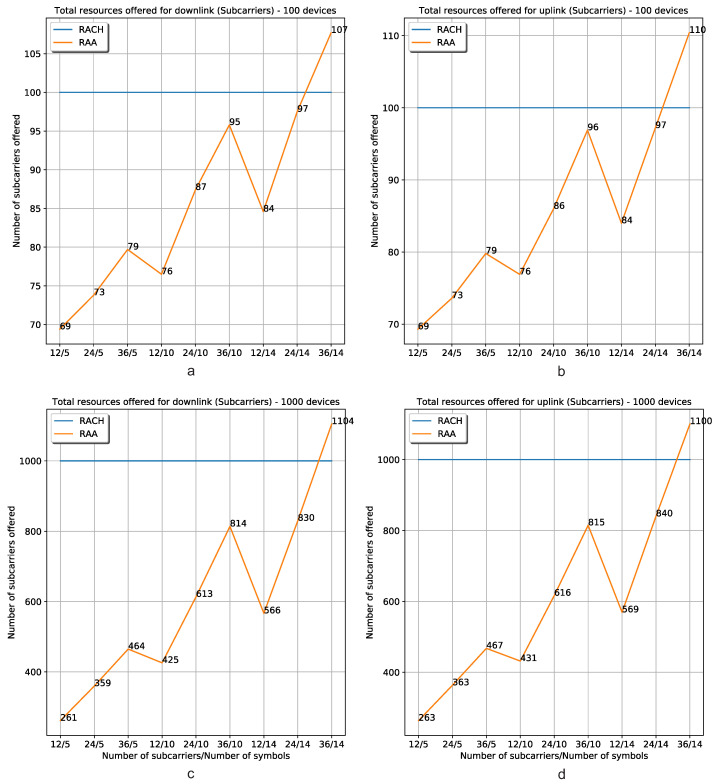
Resource allocation (number of allocated subcarriers) in the (**a**) downlink and the (**b**) uplink when there are 100 devices registered, and resource allocation in the (**c**) downlink and (**d**) uplink when there are 1000 devices connected to the mobile network using the classic resource allocation procedure (the result of RACH execution) and the RAA approach.

**Figure 11 sensors-20-05485-f011:**
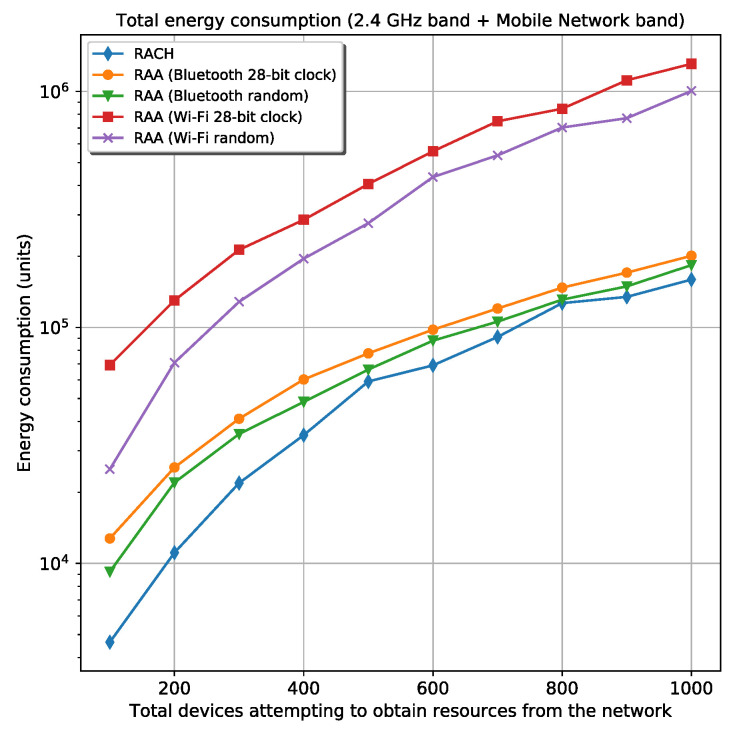
Total energy spent for all devices registration in all bands when there are 1000 connected devices to the mobile network before the new requesters start attempting to obtain resources from the network for the classic RACH and four different RAA procedures.

**Figure 12 sensors-20-05485-f012:**
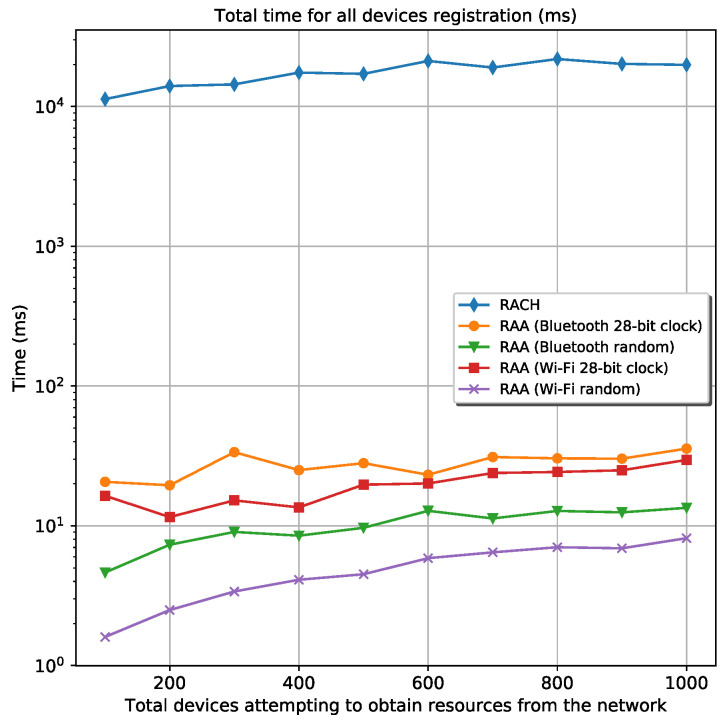
Elapsed time for all devices registration in all bands when there are 1000 connected devices to the mobile network before the new requesters start attempting to obtain resources from the network for the classic RACH and four different RAA procedures.

**Table 1 sensors-20-05485-t001:** Functionalities of the gNB, the requester, and the relay.

**gNB**	Sending the SIB1 with the access probability for ACB. Scanning incoming *Random-Access Requests*. Scanning incoming messages from relays with the resource information of the devices that want to connect to the mobile network. Registering *Random-Access-Radio Network Temporary Identifier* (RA-RNTI) and assigning *Temporary/Cell-RNTI* (T/C-RNTI) when *Random-Access Requests* arrive. Sending back RACH responses. Checking the *Time-To-Live* (TTL) of every device. Scanning incoming *RRC Connection Request* messages. Allocating resources. Sending back *RRC Connection Setup* messages.
**Requester**	Scanning incoming SIB1. Executing the ACB procedure. Performing the *Random-Access* procedure. Sending discovery messages to nearby relays. Generating the discovery frequencies used for the D2D communication. Stopping the *Random-Access* procedure and the device discovery when an acknowledge message from nearby relays or a RACH response arrives from the gNB. Starting again the *Random-Access* and the proposed RAA procedures when the *ra-ResponseWindow* expires. Sending *RRC Connection Request* message for resource allocation when RACH response arrives. Applying *back-off* when there is not RACH response after *ra-ResponseWindow*. Scanning incoming *RRC Connection Setup* message. Applying a different *back-off* when the inquiring interval ends, to avoid the same number of collisions than before in the 2.4 GHz band.
**Relay**	Scanning incoming discovery messages. Generating the frequencies used for the discovery message scanning. Sending back a response for the discovery messages. Forwarding the discovery messages to the gNB in its mobile network resources.

**Table 2 sensors-20-05485-t002:** Parameters used by the gNB in the simulation.

Parameter	Value
Access probability sent by gNB	Real range [0.2, 0.8]
SIB1 periodicity	5 ms
Numerology	*Subcarrier Spacing* (SCS) = 15 kHz, Time slot = 66.67 μs
TTL	*sl1* = 66.67 μs
RRC interval (wait for RRC request)	*sl1* = 66.67 μs
Resource allocation capacity	*∞*
2.4 GHz slices (Groups of D2D)	52 (*Bluetooth*) or 39 (*Wi-Fi*)
gNB transmission power	24 dBm
T_ACB_	4 s
T_barring_	[0.7 + 0.6 *U*[0, 1)]*T_ACB_*
Simulation time resolution	31.25 μs

**Table 3 sensors-20-05485-t003:** Parameters used by the devices in the simulation.

Parameter	Value
RBs requested by devices	Integer range [1 (12 subcarriers), 3 (36 subcarriers)]
Maximum number of symbols requested by devices	14
Transmission power	8 dBm (*Bluetooth*) and 20 dBm (*Wi-Fi*)
Coverage area	*Bluetooth* radius = 50 m, *Wi-Fi* radius = 100 m
D2D frequency used for inquiring	28-bit clock frequency generator (from *Bluetooth*) or random (32 frequencies)
D2D frequency used for scanning	28-bit clock (from *Bluetooth*)
Inquiring slot	312.5 μs
Interval where devices turn on	Real range [0 ms, 15 ms] (random)
Simulation time resolution	31.25 μs

**Table 4 sensors-20-05485-t004:** Metrics used in the simulation.

Metric	Description
Energy consumption	Represents the number of transmissions multiplied by the transmission power of the wireless technology used for the transmission. Then, it is normalized by the transmission power of *Bluetooth* (8 dBm = 6.3 mW). This is a dimensionless quantity.
Collisions	Number of collisions experienced: Number of transmissions in the same channel and at the same time. This is a dimensionless quantity.
Time	Total time for device’s registration in the mobile network. Given in milliseconds.

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
