# Peer review of "Random-Access Accelerator (RAA): A Framework to Speed Up the Random-Access Procedure in 5G New Radio for IoT mMTC by Enabling Device-To-Device Communications"

_sensors, 2020, doi:10.3390/s20195485_

Round 1
Reviewer 1 Report
In this paper, a methodology named as 'Framework' is proposed to speed up network access in massive Machine Type Communication (mMTC) into 5G cellular networks. The proposed 'Framework' approach exploits Device-To-Device (D2D) communications to gain access in the network. The authors have tried both WiFi and Bluetooth for D2D communications. In comparison to the conventional Random-Access procedure, the authors show an acceleration in the registration procedure of 99%, a freed space of the allocated spectrum until 74%.
In general the paper's technical contributions are acceptable to the journal for publications. The authors have also demonstrated a good organization style and presentation of the results. Therefore I recommend its acceptance.
- Some sections in the paper are too short and there are many small sections. They can merged with the previous ones.
- The related work section should be updated taking into account the differences with the state of art.
- The main contributions of the paper should be summarized after related work.
- The sentence on page 17 line 475 need to be rephrased.
- The figures fonts should be normalized as some have higher and some have low font sizes
- Why do the authors use the name Framework. A better naming convention can be good as the name is generic and cannot distinguish the proposed framework in better manner.
- A reference should be given for the Simpy module used in simulations.
- The functionalities of gNodeB, requester and relay on page 8 and 9 can be given in a tabular format.
- The metrics used in simulations such as energy consumption, collisions, etc and their corresponding units (Joules, etc) should be given a separate formula or appropriately defined at the beginning of the simulation. For example what does units mean in energy consumption? Is it in Joules?
- Conclusion section should be improved a little bit and show the simulation results and analysis results.
Reviewer 2 Report
This work proposes a Framework to speed up network access in massive Machine Type Communication (mMTC). Their work exploits Device-To-Device (D2D) communications where devices with already assigned resources act like relays for the rest of devices trying to gain access in the network. To improve this work, the author should address the following points: 1. The paper presentabilitty: a. The motivation: the motivation of this paper is not well written in the introduction part. The most relevant reference which is provided in introduction is reference 6 and 7. These references are not enough to show the motivation of this work. Try to introduce few issues which are are found in reference 22, 23, 24 within the introduction b. Organization Section 2 and section 3 namely NR Random-Access Procedure and Random-Access Procedure Constraints could be subsections of one section to show the preliminaries of the proposed approach. it does not look good to have these two sections then later have the related work. Related work should follow (when possible) the introduction or contribution subsection. 2. There are several english typos and phrase reconstruction that need to be addressed. For instance in line 184, the authors cited "In [23] are considered various resources allocation...", they can improve it is in [23], the authors considered.....! The paper requires a thorough proof reading 3. The last section of simulation is very well written and extensive but the figures/plot have different sizes and different fonts, thus it does not look good, the authors should work on it and improve it.Author Response
Please see the attachment

Round 2
Reviewer 2 Report
The comments and suggestions were addressed accordingly